# Carbapenem-Resistant *Pseudomonas aeruginosa* Bacteremia, through a Six-Year Infection Control Program in a Hospital

**DOI:** 10.3390/microorganisms11051315

**Published:** 2023-05-17

**Authors:** Amalia Papanikolopoulou, Panagiotis Gargalianos-Kakolyris, Athina Stoupis, Nikos Moussas, Anastasia Pangalis, Kalliopi Theodoridou, Genovefa Chronopoulou, Nikos Pantazis, Maria Kantzanou, Helena C. Maltezou, Athanasios Tsakris

**Affiliations:** 1Clinical Pharmacology Department, Athens Medical Center, 5-7 Distomou Str., 15125 Marousi, Greece; amaliapapaniko@yahoo.com; 2Clinical Infectious Diseases Department, Athens Medical Center, 1 Delfon Str., 15125 Marousi, Greece; pagargalianos@gmail.com (P.G.-K.); astoupis@otenet.gr (A.S.); drnikolaosmoussas@gmail.com (N.M.); 3Biopathology Department, Athens Medical Center, 5-7 Distomou Str., 15125 Marousi, Greece; a.pangalis@iatriko.gr (A.P.); g.chronopoulou@iatriko.gr (G.C.); 4Department of Microbiology, School of Medicine, National and Kapodistrian University of Athens, 75 Mikras Asias Str., 11527 Athens, Greece; lmktheo@yahoo.co (K.T.); mkatzan@med.uoa.gr (M.K.); atsakris@med.uoa.gr (A.T.); 5Department of Microbiology, Andreas Syggros Hospital for Skin and Venereal Diseases, National and Kapodistrian University of Athens, 11527 Athens, Greece; 6Department of Hygiene, Epidemiology and Medical Statistics, Faculty of Medicine, School of Health Sciences, National and Kapodistrian University of Athens, 75 Mikras Asias Str., 11527 Athens, Greece; npantaz@med.uoa.gr; 7Directorate of Research, Studies and Documentation, National Public Health Organization, 3-5 Agrafon Str., 15123 Athens, Greece

**Keywords:** carbapenem-resistant *Pseudomonas aeruginosa* bacteremia, antibiotic consumption, infection control interventions, hospital

## Abstract

Background: Carbapenem-resistant *Pseudomonas aeruginosa* (CRPA) is a life-threatening healthcare-associated infection affecting especially patients with immunosuppression and comorbidities. We investigated the association between the incidence of CRPA bacteremia, antibiotic consumption, and infection control measures in a hospital during 2013–2018. Methods: We prospectively recorded the incidence of CRPA bacteremia, antibiotic consumption, use of hand-hygiene solutions, and isolation rates of multidrug-resistant (MDR) carrier patients. Findings: The consumption of colistin, aminoglycosides, and third-generation cephalosporins decreased significantly in the total hospital and its divisions (*p*-value < 0.001 for all comparisons) while the consumption of carbapenems decreased significantly in the adults ICU (*p*-value = 0.025). In addition, the incidence of CRPA significantly decreased in the total hospital clinics and departments (*p*-values = 0.027 and 0.042, respectively) and in adults clinics and departments (*p*-values = 0.031 and 0.051, respectively), while in the adults ICU, the incidence remained unchanged. Increased isolation rates of MDR carrier patients, even two months before, significantly correlated with decreased incidence of CRPA bacteremia (IRR: 0.20, 95% CI: 0.05–0.73, *p*-value = 0.015) in the adults ICU. Interestingly, when the use of hand-hygiene solutions (alcohol and/or scrub) increased, the consumption of advanced, nonadvanced, and all antibiotics decreased significantly. Conclusion: In our hospital, multimodal infection control interventions resulted in a significant reduction of CRPA bacteremia, mostly due to the reduction of all classes of antibiotics.

## 1. Introduction

*Pseudomonas aeruginosa* is a significant cause of healthcare-associated infections associated with high morbidity and mortality in many groups of patients, especially those with immunosuppression and several comorbidities [1,2]. High virulence stems from the ability to escape the host immune response due to genomic fluidity, along with antibiotic resistance determinants and the formation of multicellular biofilms [3,4]. Successful antivirulence therapy requires the overcoming of multiple intrinsic and acquired resistant mechanisms, which proves to be particularly difficult to achieve with standard antibiotic therapy [5].

The 2017 World Health Organization (WHO) global priority list of pathogens ranks carbapenem-resistant *P. aeruginosa* (CRPA) in the highest priority category [6]. The reported rates of carbapenem resistance seem to be considerably higher for nonfermenter pathogens, especially for *P. aeruginosa* (frequently > 60%) than for fermenter ones (frequently < 10%) worldwide, in all types of infection [7]. In Greece, CRPA strains concurrently producing VIM and KPC emerged during the 1980s, right after the launch of imipenem and meropenem as therapeutic agents against *P. aeruginosa* infections [8].

Antimicrobial stewardship programs (ASPs) have been implemented in many countries in order to control *P. aeruginosa* resistance [9,10,11]. Recommendations for infection prevention and control measures have been issued in order to reduce the incidence of carbapenem-resistant pathogens [12]. In addition, many interventions to improve antibiotic-prescribing practices in the hospital setting have been investigated, especially for carbapenems [13,14].

In our previous published work [15] we focused on studying infection control interventions and outcomes in a hospital with an active, education-based infection control program. The most significant interventions were the increased consumption of hand disinfectant solutions and the increased isolation of multidrug-resistant (MDR) carrier patients, which indicated adherence to the program [15]. In addition, the most important outcome was the reduced consumption of advanced antibiotics, including carbapenems, which indicated adherence to the restricted formulary [15]. Overall, the impact of this six-year program on the incidence of total CR Gram-negative bacteremia resulted in a decreasing, though not statistically significant, trend in the hospital [15].

The current study aimed to evaluate the incidence of CRPA bacteremia during the implementation of this ASP focusing on the restrictive-prescribing antibiotic surveillance.

## 2. Materials and Methods

### 2.1. Study Design

The study was conducted prospectively in a 300-bed tertiary-care hospital in Athens, Greece during a six-year period (January 2013 through December 2018). The hospital has: 1. one adults clinic [with internal medicine, surgery, hematology, and oncology departments and one intensive care unit (ICU)]; 2. one obstetrics and gynecology clinic (with one neonatal ICU); and 3. one pediatric clinic (with one pediatric ICU).

### 2.2. Interventions

The following interventions were applied: 1. surveillance of CRPA among other CR pathogens; 2. formulary restriction and preauthorization for prescription of advanced antibiotics; 3. promotion of hand hygiene before and after providing healthcare to patients (soap, scrub disinfectant solutions with chlorhexidine, and alcohol 70% disinfectant solutions with chlorhexidine); and 4. multibody-site colonization screening cultures (pharyngeal, axillary-rectal, and nasal) and isolation of MDR carrier patients. The program was evaluated on a monthly basis by applying prospective audits followed by feedback interventions.

### 2.3. Data Collection and Outcomes

Data were collected prospectively on a monthly basis. The following outcomes were estimated on a monthly basis: 1. incidence of CRPA bacteremia; 2. antibiotic consumption; 3. consumption of hand disinfectant solutions; and 4. isolation rate of MDR carrier patients.

### 2.4. Detection of Bacteremia and Microbial Resistance

Bacteremia was detected through Gram stains and blood cultures. The automated VITEK 2 system (Biomerieux, Marcy-l’ Etoile, France) was used for the isolation, identification, and antibiotic susceptibility testing. The CLSI breakpoints were applied. The assay (Kirby–Bauer test, MIC semiautomated testing, and E-test) used to determine antibiotic susceptibility was recorded. During the study period, no changes were made in laboratory diagnostic procedures.

### 2.5. Definitions

Bacteremia was defined as a laboratory-confirmed bloodstream infection, either primary (not related to an infection at another body site) or secondary (thought to be seeded from a site-specific infection at another body site) [16]. A new episode of bacteremia due to a different pathogen strain or due to the same pathogen strain but with a different phenotype of resistance within a month was considered a new episode of bacteremia [15]. As we have already described in our previous work, the incidence of the total CR Gram-negative bacteremia is the sum of the incidence of CR *K. pneumoniae*, *P. aeruginosa*, and *A. baumanii*, bacteremia per 1000 patient-days [15]. The 2018 version of the WHO ATC/DDD index was used to present data on antibiotic consumption [numbers of defined daily doses (DDD) per 100 patient-days] [15]. The following antibiotics were defined as advanced antibiotics: carbapenems, colistin, tigecycline, fosfomycin, linezolid, daptomycin, ceftaroline, ceftazidime-avibactam, and ceftolozane-tazobactam [15]. Hand hygiene concerns the use of alcohol 70% disinfectant solutions with chlorhexidine, scrub disinfectant solutions with chlorhexidine, and/or simple soap, and consumption was expressed as L per 1000 patient-days [15]. The isolation rate of MDR-carrier patients was expressed as a percentage of isolated patients per admission [15].

### 2.6. Statistical Analysis

We investigated time trends in the intervention and outcome variables during the six-year study period. Time since January 2013 (the beginning of the study) was the independent variable in the regression models and entered through appropriate restricted cubic splines. In each case, the dependent variable was the variable under investigation (intervention or outcome). Fourier series terms of time (1st and 2nd order) were also entered into the models to capture potential seasonality effects. In all cases, standard errors (SE) and the corresponding 95% confidence intervals (CI) were estimated using the robust (sandwich) variance estimator to adjust potential violations of models’ assumptions [17]. Estimated values for the beginning and end of the study period and the corresponding 95% CIs were estimated through a simplification of the models. Spline time terms were replaced by a single linear time trend or two piecewise linear terms to capture the average long-term trend [17]. Estimates for annual change rates (along with 95% CI and *p*-values) are given based on the results from the linear or piecewise linear models with one-knot placement. The choice of the latter was made between the linear or piecewise linear model based on checks of all possible points (excluding the first and last year of the study), maximizing the likelihood (or equivalent minimizing the square error), as well as tests of the statistical significance of the difference in slopes before and after the optimal point of change (knot) of the slope. A linear-regression model was applied for the consumption of antibiotics and the consumption of disinfectants [17]. Poisson regression models were used with the number of cases as a dependent variable and the appropriate number of patient-days as an offset after logarithmic transformation, for cases where the outcome of interest was bacteremia rates [17]. Binomial regression models were used with the number of cases as the dependent variable and the appropriate number of hospitalizations as the binomial denominator, for cases in which the percentage over the total number of hospitalizations was the outcome of interest (isolations) [17]. We introduced appropriate independent variables into the models to investigate the associations between outcomes and interventions. The effects of the independent variables were initially tested separately for current (“month 0”) and lagged values (months −1, −2, and −3). If the effects were statistically significant (*p* value < 0.05) or indicative (0.05 < *p* value < 0.10) for more than one case (e.g., in month 0 and in month −1) and association direction was the same (e.g., positive for both), the average value was used as the independent variable. In cases where the direction of the association was different (e.g., positive for “month 0” and negative for “month −1”), the results of the respective models are presented separately. The *p* values have not been adjusted for multiple testing [17]. The Stata version 14.2 (Stata Corp., College Station, TX, USA) was used for analyses.

## 3. Results

From January 2013 through December 2018, 95,228 admissions were recorded in the hospital. Regarding bacteremia, 1671 (7.58%) cultures were positive out of a total number of 22,044 blood cultures performed during the same time period [15]. The incidence of bacteremia was estimated at 4.10/1000 patient-days for total bacteremia, 0.24/1000 patient-days for total CR Gram-negative bacteremia, and 0.10/1000 patient-days for CRPA. The incidences of CRPA bacteremia in the total hospital clinics and the adults ICU are displayed in Figure 1.

The time-trend analyses of different bacteremia incidences are shown in Table 1. The incidence of total bacteremia significantly increased in all hospital clinics and departments (*p*-value = < 0.001), in the adults clinic (*p*-value = 0.001) and departments (*p*-value = 0.004), and in the adults ICU (*p*-value = < 0.001), which is consistent with the increased number of admissions and blood cultures during the respective time period. However, the incidence of CRPA bacteremia decreased significantly in the total hospital clinics and departments (*p*-values = 0.027 and 0.042, respectively) and in the adults clinic and departments (*p*-values = 0.031 and 0.051, respectively). Only in the adults ICU did the incidence remain unchanged during the six-year program (0.9/1000 patient-days).

The time-trend analysis of antibiotic consumption per different antibiotic classes is shown in Table 2. Fluoroquinolones decreased significantly in the first year of the study in the total hospital and its divisions (*p*-value < 0.001 for all comparisons), while colistin, aminoglycosides, and third-generation cephalosporins had an overall 6-year decrease (*p*-value < 0.001 for all comparisons). Carbapenems decreased significantly in the total hospital clinics and in adults ICU (*p*-values = 0.008 and 0.025, respectively).

Table 3 shows the correlation between the incidence of CRPA bacteremia and the consumption of antibiotics. Not only the consumption of antibiotics used to treat CRPA infections such as colistin, fosfomycin, and aminoglycosides but also the consumption of nonadvanced antibiotics and all antibiotics correlated with increased incidence of CRPA bacteremia in the total hospital departments (*p*-values = 0.025 and 0.042 for both comparisons). For monobactams, the correlation is positive only for the current month (*p*-value = 0.078) in the adults clinic department and is inverted if the consumption is reported two and three months earlier, which is then associated with a decreased incidence of CRKP bacteremia (*p*-value = 0.008 and 0.003 for both comparisons). Interestingly, in the adults ICU, the correlation is negative for carbapenems (*p*-value = 0.011), aminoglycosides (*p*-value = 0.003), and fosfomycin (*p*-value = 0.011), and positive for monobactams three months earlier (*p*-value = 0.015), and for nonadvanced antibiotics two months earlier (*p*-value = 0.030).

Table 4 shows the correlation between CRPA bacteremia and infection control interventions. For hand-hygiene solutions, every increase in the consumption of scrub disinfectant solutions and all disinfectant solutions significantly correlated with decreased incidence of total bacteremia in all hospital departments (*p*-values = 0.011 and 0.020, respectively). Especially in the Adults ICU, an increase in the consumption of alcohol disinfectant solutions in the current month indicatively correlated with decreased incidence of CRPA bacteremia (*p*-value = 0.091). Regarding the isolation rate of patients with MDR pathogens, every increase two months earlier correlated with an increased incidence of CRPA bacteremia in the total hospital clinics and departments (*p*-values = 0.089 and <0.001, respectively) and in the adults clinics and departments (*p*-values = 0.033 and 0.004, respectively). On the contrary, in the adults ICU, every increase in the isolation rate of patients with MDR pathogens two months earlier correlated with a decreased incidence of CRKP bacteremia (*p*-value = 0.015).

Table 5 shows the correlation between the consumption of antibiotics and infection control interventions. Every increase in the isolation rate of patients carrying MDR pathogens three months earlier was associated with increased consumption of advanced antibiotics in all hospital departments and adults clinic departments (*p*-values = < 0.001 and 0.005, respectively). In addition, every increase in isolation rates in the current month or one, two, or three months earlier was associated with increased consumption of all antibiotics (*p*-values = 0.002 and 0.042, respectively). For hand-hygiene solutions, every increase in alcohol, scrub, or all hand disinfectant solutions, correlated with a decreased consumption of advanced antibiotics in the total hospital (*p*-values = 0.019, 0.004, and 0.002, respectively) and all its divisions (Table 5). The same negative correlation was repeated for nonadvanced antibiotics and all antibiotics in the total hospital departments and adults clinic departments with different post-time effects (Table 5). Interestingly, in the total hospital, including ICUs, every increase in alcohol, scrub, or all hand disinfectant solutions correlated with increased consumption of nonadvanced antibiotics and all antibiotics with different post-time effects (Table 5).

## 4. Discussion

In this six-year ASP in a 300-bed tertiary-care hospital in Athens, Greece, we aimed to prevent and control CR Gram-negative bacteremia, and in particular CRPA. Four-component multimodal interventions have been implemented in the hospital and consisted of surveillance of microbial resistance, preauthorization for prescribing advanced antibiotics, screening for and isolation of MDR carrier patients, and promotion of hand hygiene. All infection control interventions significantly improved during the study period. Regarding microbiological outcomes such as the incidence of CRPA bacteremia, we recorded a significant reduction in the total hospital and all its divisions separately except in the adults ICU, where the incidence remained unchanged. This particular result is in relevance to the significant reduction, not only of carbapenems, but also of other antibiotic classes like colistin, aminoglycosides, and third-generation cephalosporins. Within the same period, there was a decreasing, but not statistically significant, trend in other CR Gram-negative pathogens, such as *K. pneumoniae* and *A. baumannii*.

The impact of ASPs on the consumption of antibiotics has been investigated in many systematic reviews and meta-analyses, both in the adult and pediatric populations [18,19]. The inclusion of multimodal interventions in routine clinical practice supported by the institution gave positive results regarding antibiotic consumption [20,21,22,23]. *P. aeruginosa* strains are known to utilize their high levels of intrinsic and acquired resistance mechanisms to escape most antibiotics [24]. From whole-genome sequencing experiments of this pathogen, exposure to different classes of antibiotics (β-lactams, fluoroquinolones, and aminoglycosides) drives stress evolution, leading to the alarming CR level in the hospital setting [25,26,27]. The success of *P. aeruginosa* in colonizing different habitats largely relies on its metabolic versatility and robustness making it difficult to eradicate this pathogen from the nosocomial environment [28]. The complexity of this microorganism causes limitations in overcoming quinolone or carbapenem resistance with standard ASPs [29,30].

In our hospital, the incidence of CRPA bacteremia was associated not only with antibiotic consumption but also with infection control measures. Regarding antibiotics, consumption of colistin, fosfomycin, and aminoglycosides along with nonadvanced antibiotics in the total, correlated significantly with increased incidence of CRPA bacteremia in the total hospital departments, indicating the stress-evolution theory [25,26]. For monobactams, the correlation is positive only for the current month, implying its use as first-line therapy, and negative if the consumption is reported two and three months earlier because of resistance emergence and the choice of other therapeutic options. Interestingly, in the adults ICU, the correlation with antibiotics is not so conclusive, implying a complex antibiotic resistance pattern [30]. There is a need for continuing intervention strategies with different drawing frequency, evaluation, and implementation than in the rest of the hospital, to achieve a statistically significant reduction of CRPA [31,32].

Our findings point out the importance of hand hygiene for the reduction of total bacteremia in the hospital. Regarding the isolation rate of patients with MDR pathogens, there was a positive correlation with CRPA bacteremia in the total hospital clinics and departments two months earlier, implying the prolonged hospitalization duration of patients with such infections. On the contrary, in the adults ICU, every increase in the isolation rate of patients with MDR pathogens two months earlier correlated with a decreased incidence of CRKP bacteremia, implying the effectiveness of such measures in this department.

In our study, we also investigated the effect of infection control interventions on antibiotic consumption. Interestingly, when the use of hand-hygiene solutions increased the consumption of advanced antibiotics, nonadvanced antibiotics, and all antibiotics decreased, implying a positive interplay and a synergistic effect with restricted formulary intervention. On the contrary, when the rates of isolation of MDR carrier patients increased, antibiotic consumption also increased.

A clear strength of our study is the prospective collection of data during a six-year program. The following limitations should be mentioned. First, as a single-center study, there may be local factors that preclude extrapolation to other centers. Second, many models have been fitted and many hypotheses have been tested since we investigated associations between multiple outcomes and many potential predictors in several clinics. Given that adjusting *p*-values for multiple testing is controversial, we decided to present unadjusted *p*-values which cannot exclude some inflation of the type I error beyond the typical 0.05 level [33].

## 5. Conclusions

Our study provides valuable insights regarding the impact of an ASP on the incidence of CRPA bacteremia in a tertiary-care hospital from 2013 to 2018. All infection control interventions significantly improved during the study period such as hand-hygiene solutions use and isolation rates of MDR-carrier patients. Restrictive-prescribing antibiotic surveillance has led to a significant reduction in the consumption of advanced antibiotics, including carbapenems. Both the interventions and the outcomes of this program resulted in a significant reduction of CRPA bacteremia in the total hospital and its divisions, except in the adults ICU, where the incidence remained unchanged. Interestingly, correlation results showed an interplay and a synergistic effect between these interventions and outcomes, where increased use of hand-hygiene solutions resulted in a decreased consumption of advanced, nonadvanced, and all antibiotics. The implementation of multimodal infection prevention and control interventions was associated with a reduction of antimicrobial resistance in CRPA pathogens, mostly due to the reduction of all classes of antibiotics and increased hand-hygiene compliance.

## Figures and Tables

**Figure 1 microorganisms-11-01315-f001:**
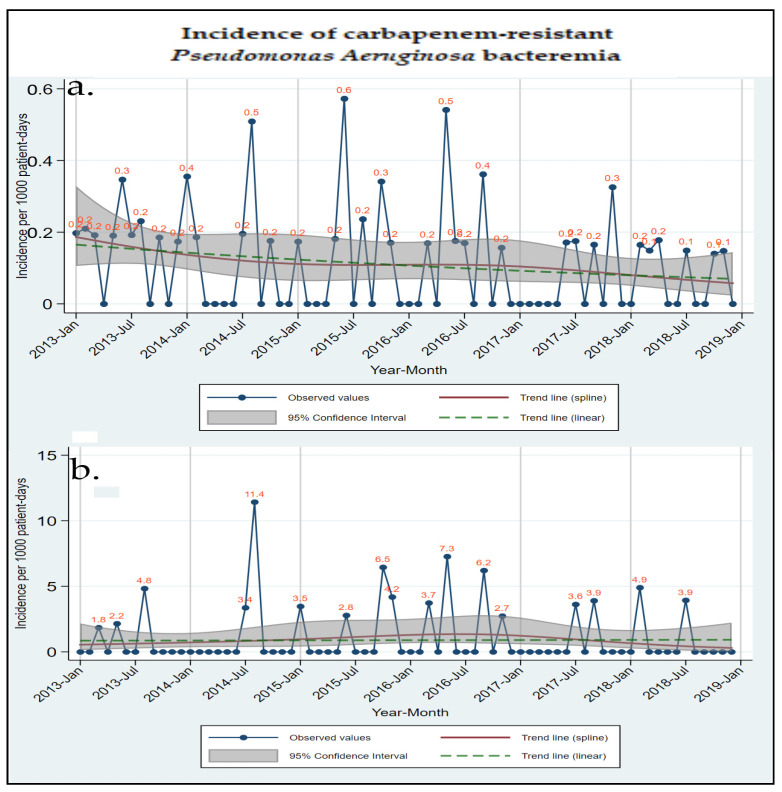
Observed values and estimated time trends for carbapenem-resistant *Pseudomonas aeruginosa* bacteremia in a hospital, from January 2013 to December 2018: (**a**) incidence in the total hospital, (**b**) incidence in the adults ICU.

**Table 1 microorganisms-11-01315-t001:** Time trend of bacteremia per 1000 patient-days in a hospital, January 2013 to December 2018.

Time Trend					
Incidence of Bacteremia/1000 Patient-Days	EVSP January 2013 (95% CI)	EVEP December 2018 (95% CI)	*p*-Value	% Relative Change/Year (95% CI)	*p*-Value
**Total Hospital Clinics**					
Total bacteremia	3.4 (3.0 to 3.8)	5.0 (4.5 to 5.5)	<0.001	2.35 (−2.15 to 7.05) up to 12/2016	0.311
				15.87 (6.67 to 25.86) after 12/2016	<0.001
Total CR Gram (−) bacteremia	0.3 (0.2 to 0.5)	0.2 (0.1 to 0.3)	0.099	−7.78 (−16.24 to 1.53)	0.099
Total CRPA bacteremia	0.2 (0.1 to 0.2)	0.1 (0.0 to 0.1)	0.027	−13.54 (−24.02 to −1.62)	0.027
**Total Hospital Departments**					
Total bacteremia	2.7 (2.4 to 3.1)	4.4 (3.5 to 5.5)	<0.001	2.37 (−1.72 to 6.63) up to 12/2017	0.260
				43.24 (10.34 to 85.95) after 12/2017	0.007
Total CR Gram (−) bacteremia	0.2 (0.1 to 0.3)	0.1 (0.1 to 0.2)	0.165	−11.12 (−24.74 to 4.97)	0.165
Total CRPA bacteremia	0.1 (0.1 to 0.2)	0.0 (0.0 to 0.1)	0.042	−22.86 (−39.94 to 0.92)	0.042
**Adults Clinic**					
Total bacteremia	4.8 (4.2 to 5.5)	6.3 (5.6 to 7.2)	0.001	−0.57 (−5.41 to 4.51) up to 11/2016	0.821
				15.50 (5.11 to 26.91) after 11/2016	0.003
Total CR Gram (−) bacteremia	0.6 (0.4 to 0.8)	0.4 (0.3 to 0.5)	0.119	−7.35 (−15.83 to 1.98)	0.119
Total CRPA bacteremia	0.3 (0.2 to 0.4)	0.1 (0.1 to 0.2)	0.031	−13.53 (−24.22 to −1.33)	0.031
**Adults Clinic Departments**					
Total bacteremia	2.9 (2.6 to 3.4)	4.1 (3.6 to 4.7)	0.004	5.91 (1.85 to 10.14)	0.004
Total CR Gram (−) bacteremia	0.3 (0.2 to 0.5)	0.2 (0.1 to 0.3)	0.205	−10.01 (−23.56 to 5.93)	0.205
Total CRPA bacteremia	0.2 (0.1 to 0.4)	0.0 (0.0 to 0.1)	0.051	−21.77 (−38.86 to 0.10)	0.051
**Adults ICU**					
Total bacteremia	18.2 (13.9 to 23.7)	32.8 (27.5 to 39.2)	<0.001	−3.81 (−15.16 to 9.07) up to 02/2016	0.545
				28.57 (14.91 to 43.85) after 02/2016	<0.001
Total CR Gram (−) bacteremia	2.5 (1.7 to 3.5)	3.3 (2.1 to 5.1)	0.392	4.91 (−5.99 to 17.07)	0.392
Total CRPA bacteremia	0.9 (0.4 to 1.8)	0.9 (0.4 to 2.1)	0.909	1.25 (−18.13 to 25.21)	0.909

ICU: intensive care unit; CR: carbapenem resistant; CRPA: carbapenem-resistant *Pseudomonas aeruginosa*; EVSP: Estimated Value Start Period, EVEP; Estimated Value End Period; CI: Confidence Interval; All estimates derived from Poisson regression models with robust standard errors, seasonality terms and linear or piecewise linear long term trend: log(N) = β_0_ + β_1_t_−_ + β_2_t_+_ + β_3_ × sin(2πt/12) + β_4_ × cos(2πt/12) + β_5_ × sin(4πt/12) + β_6_ × cos(4πt/12) + log(patient-days) with N being the number of cases and t being time since study start in months (t_−_ and t_+_ piecewise linear time terms; when piecewise linear long term trend was not required a single time term was used). % Relative changes/year derived as [exp(12 × β_1,2_) − 1] × 100%.

**Table 2 microorganisms-11-01315-t002:** Time trend of antibiotic consumption (DDDs/100 patient-days) in a hospital, January 2013 to December 2018.

Time Trend					
Antibiotic Consumption DDDs/100 Patient-Days	EVSP January 2013 (95% CI)	EVEP December 2018 (95% CI)	*p*-Value	% Relative Change/Year (95% CI)	*p*-Value
**Total Hospital Clinics**					
Fluoroquinolones	10.2 (9.6 to 10.7)	10.6 (10.2 to 11.0)	0.176	−2.96 (−3.72 to −2.20) up to 2/2014	<0.001
				0.69 (0.56 to 0.82) after 2/2014	<0.001
Colistin	2.7 (2.4 to 3.0)	1.4 (1.2 to 1.6)	<0.001	−0.92 (−1.14 to −0.70) up to 08/2014	<0.001
				0.04 (−0.04 to 0.11) after 08/2014	0.329
Aminoglycosides	6.5 (6.3 to 6.7)	4.3 (4.1 to 4.5)	<0.001	−0.36 (−0.42 to −0.31)	<0.001
3rd generation cephalosporins	16.1 (14.6 to 17.6)	7.3 (6.9 to 7.7)	<0.001	−12.42 (−14.11 to −10.73) up to 12/2013	<0.001
				0.51 (0.37 to 0.66) after 12/2013	<0.001
Carbapenems	9.3 (8.3 to 10.4)	7.8 (7.3 to 8.2)	0.008	−2.30 (−3.06 to −1.53) up to 6/2014	<0.001
				0.37 (0.23 to 0.51) after 6/2014	<0.001
**Total Hospital Departments**					
Fluoroquinolones	11.0 (10.3 to 11.7)	11.6 (11.1 to 12.0)	0.159	−3.13 (−4.08 to −2.18) up to 1/2014	<0.001
				0.76 (0.60 to 0.91) after 1/2014	<0.001
Colistin	1.6 (1.3 to 1.9)	0.9 (0.8 to 1.1)	<0.001	−0.41 (−0.63 to −0.19) up to 08/2014	<0.001
				−0.01 (−0.07 to 0.06) after 08/2014	0.816
Aminoglycosides	7.3 (6.8 to 7.7)	4.5 (4.3 to 4.7)	<0.001	−1.00 (−1.35 to −0.65) up to 7/2014	<0.001
				−0.29 (−0.36 to −0.21) after 7/2014	<0.001
3rd generation cephalosporins	18 (1.3 to 1.9)	7.6 (0.8 to 1.1)	<0.001	−14.05 (−15.87 to −12.22) up to 12/2013	<0.001
				0.49 (−0.34 to 0.64) after 12/2013	<0.001
Carbapenems	8.1 (7.2 to 9.0)	7.6 (7.0 to 8.2)	0.336	−1.83 (−2.53 to −1.14) up to 7/2014	<0.001
				0.50 (0.32 to 0.68) after 7/2014	<0.001
**Adults Clinic**					
Fluoroquinolones	16.2 (14.9 to 17.4)	17.8 (17.1 to 17.4)	0.024	−4.00 (−5.60 to −2.39) up to 1/2014	<0.001
				1.14 (0.90 to 1.38) after 1/2014	<0.001
Colistin	4.2 (3.7 to 4.8)	2.4 (2.1 to 2.6)	<0.001	−1.30 (−1.70 to −0.89) up to 08/2014	<0.001
				0.04 (−0.09 to 0.16) after 08/2014	0.565
Aminoglycosides	9.4 (9.1 to 9.7)	6.1 (5.9 to 6.4)	<0.001	−0.55 (−0.63 to −0.47)	<0.001
3rd generation cephalosporins	23.7 (21.6 to 25.9)	8.1 (7.4 to 8.7)	<0.001	−17.83 (−20.02 to −15.64) up to 1/2014	<0.001
				0.44 (0.21 to 0.67) after 1/2014	<0.001
Carbapenems	13.7 (11.7 to 15.7)	12.1 (11.3 to 12.9)	0.123	−2.34 (−3.82 to −0.86) up to 7/2014	0.002
				0.42 (0.17 to 0.67) after 7/2014	0.001
**Adults Clinic Departments**					
Fluoroquinolones	17.1 (14.6 to 19.6)	17.9 (17.2 to 18.6)	0.539	−5.08 (−7.87 to −2.29) up to 1/2014	0.001
				1.19 (0.93 to 1.46) after 1/2014	<0.001
Colistin	2.2 (1.7 to 2.7)	1.4 (1.2 to 1.6)	0.004	−0.46 (−0.84 to −0.08) up to 08/2014	0.018
				−0.02 (−0.12 to 0.08) after 08/2014	0.680
Aminoglycosides	10.6 (10.3 to 10.9)	6.2 (5.9 to 6.5)	<0.001	−1.73 (−2.03 to −1.42) up to 5/2014	<0.001
				−0.45 (−0.55 to −0.36) after 5/2014	<0.001
3rd generation cephalosporins	26.2 (22.8 to 29.7)	7.6 (6.9 to 8.2)	<0.001	−20.34 (−23.82 to −16.86) up to 1/2014	<0.001
				0.33 (0.09 to 0.58) after 1/2014	0.008
Carbapenems	11.2 (9.2 to 13.2)	10.9 (10.0 to 11.7)	0.776	−1.82 (−3.25 to 0.40) up to 8/2014	0.013
				0.60 (0.32 to 0.87) after 8/2014	<0.001
**Adults ICU**					
Fluoroquinolones	20.7 (17.0 to 24.4)	18.6 (14.9 to 22.4)	0.494	−0.36 (−1.40 to 0.68)	0.494
Colistin	11.6 (8.2 to 15.1)	12.5 (9.5 to 15.5)	0.681	−1.63 (−3.20 to 0.06) up to 01/2017	0.042
				7.27 (3.12 to 11.42) after 01/2017	0.001
Aminoglycosides	9.3 (7.0 to 11.5)	7.1 (4.6 to 9.6)	0.289	−0.36 (−1.04 to −0.31)	0.289
3rd generation cephalosporins	15.1 (10.3 to 19.9)	21.3 (17.7 to 24.9)	0.036	−8.59 (−13.33 to −3.84) up to 4/2014	0.001
				3.63 (2.44 to 4.81) after 4/2014	<0.001
Carbapenems	48.5 (41.0 to 55.9)	65.2 (51.7 to 78.8)	0.025	−1.78 (−4.39 to 0.83) up to 3/2017	0.177
				13.81 (4.08 to 23.55) after 3/2017	0.006

DDD: Daily Dose Defined, EVSP: Estimated Value Start Period; EVEP: Estimated Value End Period; CI: Confidence Interval; All estimates derived from linear regression models with robust standard errors, seasonality terms, and piecewise linear long-term trend: E[Y] = β_0_ + β_1_t_−_ + β_2_t_+_ + β_3_ × sin(2πt/12) + β_4_ × cos(2πt/12) + β_5_ × sin(4πt/12) + β_6_ × cos(4πt/12) with E[Y] being the expected consumption value and t being time since study start in months (t_−_ and t_+_ piecewise linear time terms). Absolute changes/year derived as β_1_,_2_ × 12.

**Table 3 microorganisms-11-01315-t003:** Incidence of total carbapenem-resistant *Pseudomonas aeruginosa* bacteremia and correlation with consumption of antibiotics, January 2013 to December 2018.

Total CRPA Bacteremia Correlation with Antibiotics								
Antibiotics (DDDs/100 Patient-Days)	Per (n)DDD	Month 0	Month −1	Month −2	Month −3	IRR	95% CI	*p*-Value
**Total Hospital Clinics**								
Penicillin total	1	◊				1.41	(0.96, 2.08)	0.076
Monobactams	0.1				◊	0.72	(0.49, 1.05)	0.084
**Total Hospital Departments**								
Aminoglycosides	1			◊		2.41	(0.94, 6.21)	0.068
Colistin	1		◊			2.42	(0.85, 6.91)	0.098
Fosfomycin	0.1		◊			1.39	(1.05, 1.85)	0.021
Non-Advanced Antibiotics	1			◊	◊	1.31	(1.03, 1.65)	0.025
All Antibiotics	10			◊		5.92	(1.06, 32.96)	0.042
**Adults Clinic**								
Monobactams	0.1			◊	◊	0.65	(0.43, 0.98)	0.041
**Adults Clinic Departments**								
Monobactams	0.1	◊				1.40	(0.96, 2.04)	0.078
Monobactams	0.1			◊		0.47	(0.27, 0.82)	0.008
Monobactams	0.1				◊	0.35	(0.17, 0.70)	0.003
Aminoglycosides	1			◊		3.63	(1.67, 7.89)	0.001
Fosfomycin	0.1		◊			1.24	(1.03, 1.48)	0.022
Nonadvanced Antibiotics	10			◊		2.33	(1.18, 4.59)	0.014
All Antibiotics	10			◊		1.82	(1.12, 2.95)	0.016
**Adults ICU**								
Monobactams	1				◊	1.61	(1.10, 2.37)	0.015
Carbapenems	10	◊				0.64	(0.46, 0.90)	0.011
Aminoglycosides	10			◊		0.25	(0.10, 0.64)	0.003
Fosfomycin	10				◊	0.18	(0.05, 0.67)	0.011
Nonadvanced Antibiotics	10		◊			1.22	(1.02, 1.45)	0.030
Nonadvanced Antibiotics	10			◊		0.83	(0.70, 0.99)	0.037

IRR: incidence rate ratio; CI: Confidence Interval; ICU: intensive care unit; CRPA: carbapenem-resistant *Pseudomonas aeruginosa*; Symbol ◊ denotes whether the association refers to the current month consumption (month 0) value, lagged values (months −1, −2, −3) or averaged values over more than one month. Incidence Rate Ratios (IRR) refer to increases in consumption denoted in the column labeled “per (n) DDD”. All estimates derived from Poisson regression models with robust standard errors, seasonality effects and spline terms of time: log(N) = β_0_ + β_1_V + β_2_S_1_(t) + β_3_S_2_(t) + β_4_S_3_(t) + β_5_ × sin(2πt/12) + β_6_ × cos(2πt/12) + β_7_ × sin(4πt/12) + β_8_ × cos(4πt/12) +log(patient-days)with N being the number of cases, t being time since study start in months, S(t) being spline terms of t and V referring to the current month covariate (month 0) value, lagged values (months −1, −2, −3) or averaged values over more than one month. Incidence Rate Ratios (IRR) derived as [exp(n × β_1_) − 1] × 100% with n given in the column labeled “per (n) DDD”.

**Table 4 microorganisms-11-01315-t004:** Incidence of different bacteremias and correlation with infection control interventions, January 2013 to December 2018.

Correlation of Bacteremias with Infection Control Interventions								
Infection Control Interventions	Per (n) Unit	Month 0	Month −1	Month −2	Month −3	IRR	95% CI	*p*-Value
**1. Total Bacteremia**								
**Total Hospital Clinics**								**n.s.**
**Total Hospital Departments**								
L of Scrub Disinfectant sol/1000 patient-days	10	◊	◊	◊		0.81	(0.69, 0.95)	0.011
L of All Hand Disinfectant sol/1000 patient-days	10		◊			0.94	(0.90, 0.99)	0.020
**Adults Clinic**								
% Isolations/Admissions	1				◊	1.04	(1.02, 1.06)	0.001
**Adults Clinic Departments**								
% Isolations/Admissions	1				◊	1.06	(1.03, 1.10)	<0.001
**Adults ICU**								
% Isolations/Admissions	10	◊				1.20	(1.03, 1.39)	0.020
**2. Total CR Gram (−) Bacteremia**								
**Total Hospital Clinics**								n.s.
**Total Hospital Departments**								
% Isolations/Admissions	1			◊		1.54	(1.23, 1.93)	<0.001
**Adults Clinic**								n.s.
**Adults Clinic Departments**								
% Isolations/Admissions	1			◊		1.25	(1.06, 1.48)	0.009
**Adults ICU**								
% Isolations/Admissions	10	◊				2.42	(1.75, 3.35)	<0.001
% Isolations/Admissions	10			◊		0.35	(0.18, 0.66)	0.001
**3. Total CRPA Bacteremia**								
**Total Hospital Clinics**								
% Isolations/Admissions	1			◊		1.19	(0.97, 1.44)	0.089
**Total Hospital Departments**								
% Isolations/Admissions	1			◊	◊	3.60	(1.90, 6.82)	<0.001
**Adults Clinic**								
% Isolations/Admissions	1			◊		1.17	(1.01, 1.35)	0.033
**Adults Clinic Departments**								
% Isolations/Admissions	1			◊		1.48	(1.13, 1.94)	0.004
**Adults ICU**								
% Isolations/Admissions	10	◊				3.70	(1.75, 7.86)	0.001
% Isolations/Admissions	10		◊			0.40	(0.14, 1.16)	0.092
% Isolations/Admissions	10			◊		0.20	(0.05, 0.73)	0.015
L of Alcohol Disinfectant sol/1000 patient-days	10	◊				0.90	(0.79, 1.02)	0.091

IRR: incidence rate ratio; CI: Confidence Interval; ICU: intensive care unit; CR: carbapenem resistant; CRPA: carbapenem-resistant *Pseudomonas aeruginosa*; n.s.: not significant; Symbol ◊ denotes whether the association refers to the current month (month 0) value, lagged values (months −1, −2, −3) or averaged values over more than one month. Incidence Rate Ratios (IRR) refer to increases denoted in the column labeled “per (n) units”. All estimates derived from Poisson regression models with robust standard errors, seasonality effects, and spline terms of time: log(N) = β_0_ + β_1_V + β_2_S_1_(t) + β_3_S_2_(t) + β_4_S_3_(t) + β_5_ × sin(2πt/12) + β_6_ × cos(2πt/12) + β_7_ × sin(4πt/12) + β_8_ × cos(4πt/12) + log(patient days) with N being the number of cases, t being time since study start in months, S(t) being spline terms of t and V referring to the current month covariate (month 0) value, lagged values (months −1, −2, −3) or averaged values over more than one month. Incidence Rate Ratios are derived as exp(n × β_1_) with n given in the column labeled “per (n)”.

**Table 5 microorganisms-11-01315-t005:** Antibiotics correlation with infection control interventions, January 2013–December 2018.

Antibiotics Correlation with Infection Control Interventions								
Infection Control Interventions	Per (n) Unit	Month 0	Month −1	Month −2	Month −3	β	95% CI	*p*-Value
**Advanced Antibiotics**								
**Total Hospital Departments**								
% Isolations/Admissions	1				◊	0.53	(0.27, 0.79)	<0.001
L of Alcohol Disinfectant sol/1000 patient-days	10	◊	◊	◊	◊	−0.57	(−1.05, −0.10)	0.019
L of Scrub Disinfectant sol/1000 patient-days	10	◊				−0.75	(−1.25, −0.25)	0.004
L of All Hand Disinfectant sol/1000 patient-days	10	◊	◊	◊	◊	−0.50	(−0.82, −0.19)	0.002
**Adults Clinic**								
% Isolations/Admissions	1				◊	0.47	(0.08, 0.85)	0.019
L of Alcohol Disinfectant sol/1000 patient-days	10	◊	◊	◊	◊	−1.63	(−2.40, −0.86)	<0.001
L of Scrub Disinfectant sol/1000 patient-days	10	◊				−0.98	(−1.89, −0.07)	0.035
L of All Hand Disinfectant sol/1000 patient-days	10	◊	◊			−1.06	(−1.63, −0.49)	<0.001
**Adults Clinic Departments**								
% Isolations/Admissions	1				◊	0.51	(0.16, 0.87)	0.005
L of Alcohol Disinfectant sol/1000 patient-days	10				◊	−0.68	(−1.35, −0.02)	0.045
L of Scrub Disinfectant sol/1000 patient-days	10	◊				−1.46	(−2.26, −0.65)	0.001
L of All Hand Disinfectant sol/1000 patient-days	10	◊				−0.81	(−1.30, −0.32)	0.002
**Adults ICU**								
% Isolations/Admissions	10	◊				9.38	(−1.60, 20.37)	0.093
**Nonadvanced Antibiotics**								
**Total Hospital Clinics**								
L of Alcohol Disinfectant sol/1000 patient-days	10	◊	◊	◊		2.99	(1.76, 4.22)	<0.001
L of All Hand Disinfectant sol/1000 patient-days	10	◊	◊	◊		1.58	(0.74, 2.42)	<0.001
**Total Hospital Departments**								
% Isolations/Admissions	1	◊				0.62	(−0.04, 1.28)	0.064
L of Alcohol Disinfectant sol/1000 patient-days	10	◊				0.97	(0.24, 1.70)	0.010
L of Scrub Disinfectant sol/1000 patient-days	10			◊	◊	−3.07	(−4.35, −1.79)	<0.001
L of All Hand Disinfectant sol/1000 patient-days	10				◊	−0.72	(−1.15, −0.29)	0.001
**Adults Clinic Departments**								
L of Alcohol Disinfectant sol/1000 patient-days	10	◊	◊			1.87	(0.42, 3.32)	0.012
L of Scrub Disinfectant sol/1000 patient-days	10	◊				−2.07	(−3.92, −0.22)	0.029
**All Antibiotics**								
**Total Hospital Clinics**								
L of Alcohol Disinfectant sol/1000 patient-days	10	◊	◊	◊		2.97	(1.29, 4.65)	0.001
L of Scrub Disinfectant sol/1000 patient-days	10			◊	◊	2.88	(0.80, 4.97)	0.007
L of All Hand Disinfectant sol/1000 patient-days	10	◊	◊	◊	◊	2.00	(0.88, 3.12)	0.001
**Total Hospital Departments**								
% Isolations/Admissions	1	◊	◊	◊	◊	2.18	(0.83, 3.52)	0.002
L of Scrub Disinfectant sol/1000 patient-days	10		◊	◊	◊	−3.14	(−5.23, −1.05)	0.004
L of All Hand Disinfectant sol/1000 patient-days	10			◊	◊	−1.02	(−1.70, −0.34)	0.004
**Adults Clinic**								
L of Scrub Disinfectant sol/1000 patient-days	10	◊				−1.89	(−3.88, 0.10)	0.062
L of All Hand Disinfectant sol/1000 patient-days	10	◊	◊			−1.86	(−3.04, −0.67)	0.003
**Adults Clinic Departments**								
% Isolations/Admissions	10	◊				1.58	(0.06, 3.10)	0.042
L of Alcohol Disinfectant sol/1000 patient-days	10	◊				−3.53	(−6.07, −0.98)	0.007
L of All Hand Disinfectant sol/1000 patient-days	10	◊				−1.81	(−3.35, −0.28)	0.021

CI: Confidence Interval; ICU: intensive care unit; CR: carbapenem resistant; CRPA: carbapenem-resistant *Pseudomonas aeruginosa;* Symbol ◊ denotes whether the association refers to the current month (month 0) value, lagged values (months −1, −2, −3), or averaged values over more than one month. β: beta coefficient refers to increases denoted in the column labeled “per (n) units”. All estimates derived from Poisson regression models with robust standard errors, seasonality effects, and spline terms of time: log(N) = β_0_ + β_1_V + β_2_S_1_(t) + β_3_S_2_(t) + β_4_S_3_(t) + β_5_ × sin(2πt/12) + β_6_ × cos(2πt/12) + β_7_ × sin(4πt/12) + β_8_ × cos(4πt/12) + log(patient-days) with N being the number of cases, t being time since study start in months, S(t) being spline terms of t and V referring to the current month covariate (month 0) value, lagged values (months −1, −2, −3) or averaged values over more than one month. Incidence Rate Ratios are derived as exp(n × β_1_) with n given in the column labeled “per (n)”.

## Data Availability

Data are available upon reasonable request.

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
