# Peer review of "Carbapenem-Resistant Pseudomonas aeruginosa Bacteremia, through a Six-Year Infection Control Program in a Hospital"

_microorganisms, 2023, doi:10.3390/microorganisms11051315_

Round 1

Reviewer 1 Report

Papanikolopoulou et al prospectively recorded incidence of carbapenem resistant Pseudomonas aeruginosa bacteremia, antibiotic consumption, use of hand hygiene solutions, and isolation rates of multidrug-resistant (MDR) carrier patients in 300-bed tertiary-care hospital in Athens. The authors have find a good correlation between the decreased consumption of colistin, aminoglycosides, and 3rd generation cephalosporins in total hospital and its divisions together with Adults ICU with the decreased incidence of total bacteremia, carbapenem resistant Gram negative bacteria and carbapenem resistant Pseudomonas aeruginosa. The study is very interesting and have great scientific value discussing an important topic which is antimicrobial stewardship program and its implication in decreasing the consumption and thus the resistance toward antimicrobials; the manuscript is well written; the results are clearly presented; however, the conclusion part needs to be rewritten. The manuscript needs also some minor corrections.

1.     Fig. 1 the name of the organism in the title should be written correctly, Pseudomonas aeruginosa rather than Pseudomonas Aeruginosa. Also, the tiles should be different (Add total hospital and adult ICU to the titles respectively). Finally, I suggest changing the numbers to Fig. 1a and Fig. 1b instead of Fig 1.1 and Fig 1.2.

2.     Table 1. Remove (MDR and N/A) from the footnote, as they were not mentioned in the table.

3.     Line 181. Change significantly the first year of the study…. To significantly in the first year of the study.

4.     Table 2. Replace the symbols έως to to.

5.     Line 200. Replace “department sand” to “departments and”.

6.     Table 5. Remove (N/S) from the footnote.

7.     The conclusion part needs to be rewritten. As it is just copied from the discussion.

Minor editing of English language required

Author Response

Comments and Suggestions for Authors

Comment 1 

Papanikolopoulou et al prospectively recorded incidence of carbapenem resistant Pseudomonas aeruginosa bacteremia, antibiotic consumption, use of hand hygiene solutions, and isolation rates of multidrug-resistant (MDR) carrier patients in 300-bed tertiary-care hospital in Athens. The authors have find a good correlation between the decreased consumption of colistin, aminoglycosides, and 3rd generation cephalosporins in total hospital and its divisions together with Adults ICU with the decreased incidence of total bacteremia, carbapenem resistant Gram negative bacteria and carbapenem resistant Pseudomonas aeruginosa. The study is very interesting and have great scientific value discussing an important topic which is antimicrobial stewardship program and its implication in decreasing the consumption and thus the resistance toward antimicrobials; the manuscript is well written; the results are clearly presented; however, the conclusion part needs to be rewritten. The manuscript needs also some minor corrections.

Fig. 1 the name of the organism in the title should be written correctly, Pseudomonas aeruginosa rather than Pseudomonas Aeruginosa. Also, the tiles should be different (Add total hospital and adult ICU to the titles respectively). Finally, I suggest changing the numbers to Fig. 1a and Fig. 1b instead of Fig 1.1 and Fig 1.2.

Answer to Comment 1

We thank the reviewer for the valuable comments to improve the article. We have rewritten the conclusion part given emphasis in the main findings of the study. Also, all corrections suggested for Figure 1 were made accordingly.

Comment 2

Table 1. Remove (MDR and N/A) from the footnote, as they were not mentioned in the table.

Answer to Comment 2

We thank the reviewer for this notice. We have removed (MDR and N/A) from the footnote, as recommended.

Comment 3

Line 181. Change significantly the first year of the study…. To significantly in the first year of the study.

Answer to Comment 3

We thank the reviewer for this notice. We corrected, as suggested.

Comment 4

Table 2. Replace the symbols έως to to.

Answer to Comment 4

We thank the reviewer for this notice. We corrected, as recommended. 

Comment 5

Line 200. Replace “department sand” to “departments and”.

Answer to Comment 5

We thank the reviewer for this notice. We corrected, as recommended.  

Comment 6

Table 5. Remove (N/S) from the footnote.

Answer to Comment 6

We thank the reviewer for this notice. We removed N/S, as recommended.  

Comment 7

The conclusion part needs to be rewritten. As it is just copied from the discussion.

Answer to Comment 7

We thank the reviewer for the valuable comments to improve the article. We have rewritten the conclusion part given emphasis to the main findings of the study.

Comment 8

Comments on the Quality of English Language

Minor editing of English language required

Answer to Comment 8

We thank the reviewer for this note. We edited the manuscript as recommended.

Reviewer 2 Report

The manuscript of Papanikolopoulou et al. describes results of an interesting extriment related to the use of antibiotics. Instead of looking for novel drugs against antibiotic-resistant strains, the authors introduced increased cleanliness in the hospital and clinics by requiring all patients and doctors to desinfect/wash hands thoroughly before contact with the patients. The data presented in the manuscript present a compelling evidence that introduction of extra cleaning steps dramatically reduces antibiotics use. The experimental data, however, show some doubts when it comes to the interpretation.

1. Figure 1 shows a statistically significant decrease in antibiotics use after introduction of the measures. The changes are small by absolute values which puts in question the "dramatic" reduction in antibiotics use after introduction of extra cleaning steps.

2. Table 1, data for CR Gram (-) and CRPA show values that are not statically valid at the 95% confidence level.

3. Table 2, the data for 3rd generation cephalosporins and carbapenems is likely grouping he same antibiotics. Therefore, it is misleading to the reader. 

The data also shows that introduction of extra cleaning steps does not make a difference in antibiotics use. Is it not contradicting the premise of the article of the antibiotics were so seldom used that there was no resistance observed in the past?

Overall, the data is compelling and shows that simple precautions like hand washing/desifections can make a dramatic difference in the appearance of drug resistant bacteria.

Author Response

Comments and Suggestions for Authors

Comment 1

The manuscript of Papanikolopoulou et al. describes results of an interesting extriment related to the use of antibiotics. Instead of looking for novel drugs against antibiotic-resistant strains, the authors introduced increased cleanliness in the hospital and clinics by requiring all patients and doctors to desinfect/wash hands thoroughly before contact with the patients. The data presented in the manuscript present compelling evidence that introduction of extra cleaning steps dramatically reduces antibiotics use. The experimental data, however, show some doubts when it comes to the interpretation.

Figure 1 shows a statistically significant decrease in antibiotics use after introduction of the measures. The changes are small by absolute values which puts in question the "dramatic" reduction in antibiotics use after introduction of extra cleaning steps.

Answer to Comment 1

We thank the reviewer for the valuable comments to improve the article. However, Figure 1 shows the observed values and estimated time trends for the incidence of carbapenem-resistant Pseudomonas Aeruginosa bacteremia during the study period and not the antibiotic consumption. The latter is shown thorhoughly in Table 2.

Big changes in antibiotic consumption (expressed in DDDs/100patient-days) in total hospital during the study period resulted in statistically reduced incidence of CRPA bacteremia (expressed per 1000 patient-days). The absolute decrease was from 0.2 to 0.1, as also shown in Figure 1a. The slope is not so sharp but is enough to show significance, which is a very important result for our hospital. Some indicative data from Table 2 are shown below:

DDDs Colistin

2.7

(2.4 to 3.0)

1.4

(1.2 to 1.6)

<0.001

DDDs Aminoglycosides

6.5

(6.3 to 6.7)

4.3

(4.1 to 4.5)

<0.001

DDDs 3rd generation cephalosporins

16.1

(14.6 to 17.6)

7.3

(6.9 to 7.7)

<0.001

Incidence Total CRPA bacteremia

0.2 (0.1 to 0.2)

0.1 (0.0 to 0.1)

0.027

Comment 2

Table 1, data for CR Gram (-) and CRPA show values that are not statically valid at the 95% confidence level.

Answer to Comment 2

We thank the reviewer for this comment. In Τables 1 and 2 the time trends of bacteremia and antibiotic consumption are shown. As shown, p-values are statistically significant (<0.05) for the estimated values at the start of the study period (EVSP) and at the end of the study period (EVEP) in total Hospital and divisions regarding CRPA, except in the Adults ICU.

Also, estimates for annual change rates (along with 95% CI and p-values) are given, based on the results from the linear or piecewise linear models. Simplified versions of the models were also fitted, where instead of splines, linear or piecewise linear terms with one knot were used to model time trends. Knot placement and choosing between the linear or piecewise linear model was based on checks of all possible points (excluding the first and last year of the study) based on maximizing the likelihood (or equivalent of minimizing the square error) as well as tests of the statistical significance of the difference in slopes before and after the optimal point of change (knot) of the slope This information was added at the statistical analysis part (Methods, page 4, lines 131-136).

The p-values for annual change rates are not always statistically significant and are not mentioned in the text.

Comment 3

Table 2, the data for 3rd generation cephalosporins and carbapenems is likely grouping the same antibiotics. Therefore, it is misleading to the reader.

Answer to Comment 3

We thank the reviewer for this comment. We decided to estimate the consumption of 3rd generation cephalosporins and carbapenems separately, in order to depict different time trends in their consumption in every division of the hospital. For example, in Adult Clinic and departments there was a statistically significant reduction in the consumption of 3rd generation cephalosporins while the consumption of carbapenems was not.

Comment 4

The data also shows that introduction of extra cleaning steps does not make a difference in antibiotics use. Is it not contradicting the premise of the article of the antibiotics were so seldom used that there was no resistance observed in the past?

Answer to Comment 4

We thank the reviewer for this comment.

From our previous published work “Papanikolopoulou A et al. Six-Year Time-Series Data on Multidrug-Resistant Bacteremia, Antibiotic Consumption, and Infection Control Interventions in a Hospital. Microb Drug Resist. 2022;28(7):806-818”, the most significant interventions were the increased consumption of hand disinfectant solutions and the increased isolation of multidrug-resistant (MDR) carrier patients, while the most important outcome was the reduced consumption of advanced antibiotics including carbapenems.

In the present study we show results regarding:

the correlation of CRPA bacteremia with antibiotics (Table 3); the correlation of CRPA bacteremia with infection control interventions (Table 4); and the correlation of Antibiotics with infection control interventions (Table 5).

In particular, as you can see in the data presented in Table 5, we show for the first time that every increase in alcohol, scrub or all hand disinfectant solutions, significantly correlated with decreased consumption of advanced antibiotics in total hospital (p-values=0.019, 0.004, and 0.002, respectively) and all its divisions. The same negative correlation was repeated for non-advanced antibiotics and all antibiotics in total Hospital Departments and Adults Clinic Departments with different post time effects (lines 257-260).

Comment 5

Overall, the data is compelling and shows that simple precautions like hand washing/desifections can make a dramatic difference in the appearance of drug resistant bacteria.

Answer to Comment 5

We thank the reviewer for this comment. Both infection control interventions and the restricted formulary separately and in conjunction can make a dramatic difference in the appearance of drug resistant bacteria.
